# Blood Serum Cytokines in Patients with Subacute Spinal Cord Injury: A Pilot Study to Search for Biomarkers of Injury Severity

**DOI:** 10.3390/brainsci11030322

**Published:** 2021-03-04

**Authors:** Sergei Ogurcov, Iliya Shulman, Ekaterina Garanina, Davran Sabirov, Irina Baichurina, Maxim Kuznetcov, Galina Masgutova, Alexander Kostennikov, Albert Rizvanov, Victoria James, Yana Mukhamedshina

**Affiliations:** 1Neurosurgical Department No. 2, Republic Clinical Hospital, 420138 Kazan, Russia; SVOgurcov@kpfu.ru (S.O.); IAShulman@kpfu.ru (I.S.); 2Clinical Research Center for Precision and Regenerative Medicine, Institute of Fundamental Medicine and Biology, Kazan Federal University, 420008 Kazan, Russia; EEGaranina@kpfu.ru (E.G.); davraniwe@gmail.com (D.S.); IABajchurina@kpfu.ru (I.B.); galina2526@gmail.com (G.M.); AleAKostennikov@kpfu.ru (A.K.); Albert.Rizvanov@kpfu.ru (A.R.); 3Department of Histology, Cytology, and Embryology, Kazan State Medical University, 420012 Kazan, Russia; qmaxksmu@yandex.ru; 4Division of Biomedical Science, School of Veterinary Medicine and Science, Faculty of Medicine and Health Sciences, University of Nottingham Biodiscovery Institute, University Park, Nottingham NG7 2RD, UK; Victoria.James@nottingham.ac.uk

**Keywords:** traumatic spinal cord injury, cytokine profile, inflammation, blood serum, clinical trial

## Abstract

*Background*. Despite considerable interest in the search for a spinal cord injury (SCI) therapy, there is a critical need to develop a panel of diagnostic biomarkers to determine injury severity. In this regard, there is a requirement for continuing research into the fundamental processes of neuroinflammatory and autoimmune reactions in SCI, identifying changes in the expression of cytokines. *Methods*. In this pilot study, an extended multiplex analysis of the cytokine profiles in the serum of patients at 2 weeks post-SCI (*n* = 28) was carried out, together with an additional assessment of neuron-specific enolase (NSE) and vascular endothelial growth factor (VEGF) levels by enzyme-linked immunosorbent assay. A total of 16 uninjured subjects were enrolled as controls. *Results*. The data obtained showed a large elevation of IFNγ (>52 fold), CCL27 (>13 fold), and CCL26 (>8 fold) 2 weeks after SCI. The levels of cytokines CXCL5, CCL11, CXCL11, IL10, TNFα, and MIF were different between patients with baseline American Spinal Injury Association Impairment Scale (AIS) grades of A or B, whilst IL2 (>2 fold) and MIP-3a (>6 fold) were significantly expressed in the cervical and thoracic regions. There was a trend towards increasing levels of NSE. However, the difference in NSE was lost when the patient set was segregated based on AIS group. *Conclusions*. Our pilot research demonstrates that serum concentrations of cytokines can be used as an affordable and rapid detection tool to accurately stratify SCI severity in patients.

## 1. Introduction

Spinal cord injury (SCI) continues to be a major and pressing problem in modern medicine [1]. To date, the treatment outcomes of patients with SCI are extremely unsatisfactory and require the development and implementation of new therapeutic protocols. In addition to the lack of effective therapeutic strategies, there is a need to develop a panel of diagnostic biomarkers to determine injury severity, which could potentially have prognostic value to aid the implementation of an optimal therapy strategy [2,3].

There are several stages in the course of SCI, each of which is characterized by certain clinical signs and pathophysiological mechanisms. The acute period of SCI is most often considered to be within 3 days from the moment of injury. This period captures the phase of primary damage, characterized by the development of pathological changes as a result of the direct impact on the tissue of a damaging factor (e.g., mechanical injury), and is quickly replaced by secondary damage [4,5]. It is in this subsequent subacute period (also referred to as the early or intermediate period) that the classical picture of secondary damage unfolds. This stage is characterized by the development of an inflammatory response, oxidative stress and excitotoxicity, which leads to the death of neurons and glial cells, the spread of ascending and descending tracts damage, and axon demyelination [4,6,7,8,9]. It should be noted that secondary injury during this period leads to more serious clinical consequences than the original primary injury [1].

Currently, there is a low rate of translation of new therapeutic approaches for SCI, with very few studies in later stage clinical trials [10,11]. A key reason for this is the fact that post-traumatic processes in animal SCI models can differ significantly from those which occur in human patients [12]. Studies on animal models show that subacute SCI promotes complex cytokine network imbalances [13,14]. In this regard, additional research aimed at deciphering the dynamics of cytokine expression in human samples after SCI is timely and an extremely relevant means of determining new markers of the post-traumatic process to uncover novel therapeutic approaches, in particular, those that may reduce the negative components of inflammatory reactions.

In addition to the search for new mechanisms upon which to base therapeutic strategies, attempts are also being made to identify biomarkers of SCI severity that could predict the outcome of the post-traumatic process [3,15]. It is considered that cerebrospinal fluid (CSF) is more representative for SCI severity assessment given its proximity to the spinal cord [11]. In this regard, most investigators focus on determining the post-traumatic levels of CSF biomarkers between SCI patients with different baseline American Spinal Injury Association Impairment Scale (AIS) grades [12,16,17,18,19]. Nonetheless, the clinical management of SCI patients does not suggest that routine CSF collection, for which there are contraindications and the risk of complications, is the best approach. The detection of markers in blood serum provides an alternative and easily accessible biofluid in which to routinely test for markers that determine SCI severity and have prognostic value.

This pilot study characterized cytokine expression at 2 weeks post-SCI in humans, establishing any associations between the injury severity or region and the concentration of inflammatory proteins, including additional assessment of neuron-specific enolase (NSE) and antigenic molecule vascular endothelial growth factor (VEGF).

## 2. Materials and Methods

### 2.1. Participants Enrollment

This study was approved by the Kazan Federal University Local Ethical Committee (Protocol No. 3, 23 March 2017). Written informed consent was obtained from each subject before blood serum was collected. Between 2017 and 2019, data were obtained from 28 patients (16 males and 7 females) suffering from subacute traumatic SCI who were admitted to the Neurosurgical Department No. 2 of the Republican Clinical Hospital (Kazan, Russia).

The following inclusion criteria were used for patients with subacute SCI in this prospective trial: (1) over 18 years of age; (2) SCI between C3 and L3 inclusive; (3) the ability to provide a valid, reliable neurological examination; and (4) classified with an AIS grade of A or B. Patients with concomitant traumatic brain injury, severe chest injuries, and intra-abdominal injuries were excluded from our study. The severity of neurological impairment with baseline AIS grade assessments were conducted by a research study neurologist experienced in these techniques and in calculating AIS. Neurological examinations were conducted at 1, 2, and 3 weeks post-injury to determine AIS conversion and possible motor and sensory score improvement.

The uninjured control group consisted of blood serum samples collected from 16 healthy able-bodied individuals, from whom we obtained consent to acquire venous blood. Uninjured subjects were recruited from the Kazan Federal University student and academic population, and hospital staff. The following inclusion criteria were used for uninjured subjects: (1) over 18 years of age; (2) without a history of SCI; (3) a lack of the main signs of inflammation; and (4) a normal complete blood count. The exclusion criteria for uninjured subjects were a previously established diagnosis of chronic inflammatory and autoimmune processes or neurological disorders, including a head trauma.

### 2.2. Sample Acquisition

Venous blood was collected (6 mL vacuum test tube, Apexlab, Moscow, Russia) via standard venipuncture at 2 weeks after injury. After 30 min of coagulation, the blood was centrifuged at 3000 rmp, divided into aliquots (300 μL), and stored at −80 °C until analysis. The blood serum samples of all participants were subjected to the same manipulations and similar storage times. To achieve this, on the day of venous blood sampling from SCI patients, venous blood sampling of uninjured individuals was also carried out in parallel.

### 2.3. Biochemical Analysis

In our study, we analyzed changes in the blood serum cytokine profile of SCI patients at 2 weeks post-injury using multiplex analysis by xMAP Luminex technology. Bio-Plex Pro™ Human Cytokine 40-plex Assay #171AK99MR2 (Bio-Rad, Hercules, CA, USA) was used, allowing simultaneous multiplex analysis of 40 (CCL21, CXCL13, CCL27, CXCL5, CCL11, CCL24, CCL26, CX3CL1, CXCL6, GMCSF, CXCL1, CXCL2, CCL1, IFNg, IL1b, IL2, IL4, IL6, IL8, IL10, IL16, CXCL10, CXCL11, MCP-1, MCP-2, MCP-3, MCP-4, CCL22, MIF, MIG, MIP-1a, MIP-1b, MIP-3a, MIP-3b, MPIF-1, CXCL16, CXCL12, CCL17, CCL25, and TNFa) human cytokines in 50 μL of the test sample. All obtained blood serum samples (28 from SCI patients and 16 from uninjured individuals) were analyzed together in the same assay. We also performed standard enzyme-linked immunosorbent assay (ELISA) on NSE (P3H 2015/2531, VectorBest, Moscow, Russia) and VEGF (P3H 2017/5974, VectorBest, Moscow, Russia) in the same serum samples. NSE and VEGF concentrations were expressed in pg/mL. The above assays were carried out strictly according to the manufacturer’s instructions and each serum sample was assayed in duplicate. All biochemical analysis was performed by investigators who were blinded to uninjured control and SCI patient groups, as well as any knowledge of the baseline AIS grade and subsequent neurological outcome.

### 2.4. Statistical Analysis

R 3.6.3 (R Foundation for Statistical Computing, Vienna, Austria) was used for data analysis. Descriptive statistics for quantitative variables are presented as mean (standard deviation) and median (interquartile range). Linear models with age as an adjusted covariate were used to estimate differences in log2-transformed expression levels of target markers. The Benjamini–Hochberg procedure was applied for multiplicity correction.

## 3. Results

### 3.1. Subject Demographics

A total of 28 subacute SCI patients were enrolled, the general demographic features and neurological status of which are shown in Table 1. Additional information concerning comorbidity and concurrent medication was collected from SCI patients and summarized in Table 2. The blood serum samples from SCI patients were acquired at 2 weeks after injury. All injured patients had stable data of AIS grade at weeks 1 to 3 post-injury (time spent in the Neurosurgical Department), and no AIS improvement was observed in the subacute period of SCI. In the 16 uninjured subjects, 8 were male and 8 were female with an average age 32.7 ± 11.5 years. The uninjured individuals did not have chronic or acute inflammatory or autoimmune conditions and did not take any medication for at least a month before blood sampling.

### 3.2. Multiplex Analysis of Blood Serum Cytokines

We performed simultaneous multiple cytokine analysis of the blood serum on 2 weeks post-SCI patients and uninjured subject samples (Appendix A, Figure 1). IFNγ, CCL27, CCL26, CXCL6, IL2, IL4, MCP-3, and MIP-3a all showed a significant elevation in the SCI patient samples (Figure 2). The greatest difference was found for IFNγ, which was more than ~52 times higher (FC 52.04 (39.68; 68.25), P_adj_ = 0.0001) in SCI patients compared to the uninjured control subjects. Large differences were also seen for CCL27 (13.10 (2.87; 59.90), P_adj_ = 0.0044) and CCL26 (8.12 (5.79; 11.39), P_adj_ = 0.0001) levels, which were both elevated 2 weeks after SCI.

MIG, IL1b, and IL10 were significantly decreased at 2 weeks post-injury compared to uninjured subject samples (Figure 2). The greatest decrease was found for IL1b (0.25 (0.18; 0.34), P_adj_ = 0.0001) in SCI patients compared to uninjured control subjects.

### 3.3. Determination of the Effect of the Spinal Cord Injury Region on Blood Serum Cytokines

To determine if the region of SCI affects blood serum cytokine levels, we compared cytokine levels when segregated based on the region of injury. SCI at cervical (C) and lumbar (L) regions had the greatest and least impact, respectively, on the cytokine profile in blood serum at 2 weeks post-injury. We observed a strong positive association between C injury and CXCL1 (2.02 (1.25; 3.26), P_adj_ = 0.0231), CXCL10 (3.13 (1.29; 7.56), P_adj_ = 0.0465), and CXCL11 (0.19 (0.05; 0.68), P_adj_ = 0.0457) concentrations (Table 3). SCI at both the C and thoracic (Th) positions produced a significant increase in IL2 and MIP-3a, as well as a decrease in MIG concentrations at 2 weeks post-injury (Figure 3, Table 3). SCI at Th and L regions, but not C, showed a positive association with CCL22 concentration (Table 3). We did not find a correlation between injury regions and CCL26, CXCL6, CCL1, IFNγ, IL10, IL1b, IL4, and MCP-3 concentrations. SCI at any region led to significant changes in above-mentioned cytokine concentrations in the blood serum at 2 weeks post-injury compared to uninjured control subjects.

### 3.4. Classifying Injury Severity with Blood Serum Cytokines

The cytokine data at 2 weeks post-SCI revealed differences existed between the AIS A and B groups of patients for CXCL5, CCL11, CXCL11, IL10, TNFα, and MIF in blood serum (Appendix A). In the AIS A patient group, an increase of CXCL5 (1.66 (1.13; 2.44), P_adj_ = 0.0299) and TNFα (1.91 (1.12; 3.26), P_adj_ = 0.0448), and a decrease in CCL11 (0.44 (0.27; 0.71), P_adj_ = 0.0055), CXCL11 (0.23 (0.08; 0.65), P_adj_ = 0.0205), and IL10 (0.44 (0.27; 0.72), P_adj_ = 0.0065) were found compared to uninjured controls (Appendix A). At the same time, elevation in MIF concentration (6.83 (1.79; 26.01), P_adj_ = 0.0193) was observed in the AIS B patient group (Appendix A).

### 3.5. NSE and VEGF Blood Serum Concentration

In addition to the panel of cytokines investigated, we also determined the levels of inflammatory protein NSE and VEGF in the blood serum by ELISA. An increase in NSE expression (P_adj_ = 0.0124) was determined at 2 weeks post-SCI compared to uninjured controls (Table 4). VEGF concentration was also increased, but not at the level of significance when compared to uninjured controls (Table 4). Despite the trend of increased NSE and VEGF remaining the same, the significance of NSE was lost when the patient set was segregated based on AIS group (Table 4). However, this potentially may be due to the reduction of numbers per AIS group compared to controls (Table 4).

## 4. Discussion

In this study, using multiplex analysis of blood serum collected from SCI patients at 2 weeks post-injury (*n* = 28), we determined the levels of 40 cytokines and 2 additional proteins, NSE and VEGF, compared to a cohort of uninjured subjects (*n* = 16). The data obtained showed a large elevation of IFNγ (>52 fold), CCL27 (>13 fold), and CCL26 (>8 fold) at 2 weeks after SCI. As IFNγ has previously been shown to be a key regulator of many cytokines, the increase in IFNγ at this stage suggests the start of a process of IFNγ-dependent cytokine modulation [20]. Th1 cells expressing IFNγ induce activation and M1 polarization of macrophages, whilst inhibiting the proliferation of Th2 cells and IL10 production, which is consistent with our data. IFNγ also modulates CCL26 synthesis in human monocytic cells, regulating inflammatory responses [21]. Upregulation of serum CCL27, as an inflammatory chemokine associated with the homing of memory T cells to sites of inflammation, has recently been identified as an indicator for multiple sclerosis [22]. Our data combined with that seen in multiple sclerosis research suggest that the inflammatory processes in SCI lead to a systemic immune response that could potentially induce autoimmune responses that closely resemble those in multiple sclerosis [23,24].

Our study found a positive association between SCI severity (AIS A) and CXCL5, TNFα, CCL11, CXCL11, and IL10 concentration at 2 weeks post-SCI. At the same time, a positive association in the MIF level was also observed in the AIS B patient group. Elevated serum levels of TNFα in subacute and chronic SCI patients were observed by Davies et al. (2007). However, this study did not draw an association with AIS grades, and unlike our study, cytokine assessments were taken at different stages of SCI spread across 2 to 52 weeks post-injury and beyond 52 weeks. Another study found a negative correlation between AIS improvement and TNFα serum levels at 9 h post-injury, but not during the subacute period [25]. Detection of IL10 concentration is often performed in CSF, but to the best of our knowledge, there have been no reports of detectable levels of serum IL10 (<1 pg/mL) obtained from either controls or SCI subjects [26]. A pilot study reported by Stein et al. (2013) showed the elevation of circulating MIF levels, but without a correlation to AIS grades or injury regions in chronic SCI patients [27]. Our results demonstrate for the first time a new set of cytokines (CXCL5, CCL11, and CXCL11) not previously described, which have a potential role in SCI pathogenesis and can act as biomarkers of injury severity. As a pilot study, our data is limited by patient number and these promising new findings remain to be verified in a larger patient dataset.

We observed some potentially interesting trends when correlating SCI region with blood serum cytokine levels. SCI in the C region had the greatest impact on the cytokine profile in the blood serum, and associations with CXCL1, CXCL10, and CXCL11 concentrations were found. IFNγ has been found to be crucial for the induction of CXCL10 and CXCL11 and the perpetuation of inflammation [28,29]. However, the level of IFNγ in our study did not correlate with injury region. Recent evidence has also indicated that serum expression of CXCL10 and CXCL11 is increased in multiple sclerosis patients and may be involved in disease pathogenesis [30,31]. CXCL1 upregulation can contribute to the maintenance of neuropathic pain and may play a critical role in stress-induced depression, which can be observed predominantly in patients with SCI occurring in the C region [32,33].

Previously, it was reported that the concentrations of biomarkers detectable within the CSF could be higher than the concentrations found within blood serum [12]. Comparative analysis of post-injury cytokine levels in both the CSF and blood serum at the same time point would demonstrate if the use of blood serum, which has huge practical advantages over CSF, could be used as an alternative fluid making fundamental changes to the management of these patients. A retrospective comparative analysis of the data we obtained from blood serum at 2 weeks post-injury with published CSF data is not possible due to differences in experimental design and patient selection. To our knowledge, the only study of 2 weeks post-injury CSF was conducted by Singh et al. (2016), which described an elevated CSF concentration of nitric oxide within motor complete paraplegia/quadriplegia SCI patients, compared to those with some spared motor function [17]. Other studies have assessed the level of some cytokines within the CSF in traumatic brain injury (TBI) patients. Of particular note, the work of Kumar et al. (2015) reconciles previously conflicting data through its concurrent evaluation of CSF and serum [34]. The study identified two TBI subpopulations with distinct CSF IL6 profiles—individuals in the high CSF temporal acute IL6 trajectory (TRAJ) had higher mean subacute serum IL1b and IL6 levels compared with the low TRAJ group.

In addition to cytokines, a small group of inflammatory proteins and trophic factors are also under investigation as markers of SCI, including NSE and VEGF. An increase in serum NSE concentration during the acute and subacute periods of SCI relative to controls has already been reported [35,36]. Our data are consistent with these previous studies, but go on to confirm that serum NSE does not appear to be a specific biomarker of injury severity when compared to AIS groups. Schwartz et al. (2020) investigated a potential association between muscle-based and serum biomarkers with pressure injury recurrence following SCI [37]. They reported that serum VEGF was significantly increased in chronic SCI patients who had never developed pressure injuries. To our knowledge, other studies did not measure serum VEGF in SCI patients, and CSF concentrations of this growth factor were not found to be of measurable levels using standard multiplex analysis [12]. Similarly, our data did not show a significant elevation of serum VEGF concentration at 2 weeks post-injury, including no difference between AIS grades.

## 5. Study Limitations

As this is a pilot study, there are several limitations that must be considered. First, the limited patient sample size. Although our cohort of 28 SCI patients (at 2 weeks post-SCI) is unevenly divided among AIS A and B severities, it is not the smallest patient cohort used for multiplex analysis of blood serum in this manner. Further testing of larger patient cohorts, especially when balanced according to SCI region and AIS severity, is warranted to ensure that the same findings are maintained. Second, uninjured controls are healthy individuals without a history of traumas or musculoskeletal disease and were not exposed to similar therapeutic regimes as the SCI patient group in this study. Further studies to validate serum biomarkers should seek to eliminate possible treatment effects where possible. Third, the single time point used in this study does not allow the tracing of the kinetics of the changes. The addition of multiple time points would strengthen the investigation and allow a more objective assessment of the changes.

## 6. Conclusions

To our knowledge, our study is the first report providing a unique description of a number of inflammatory cytokines at 2 weeks post-SCI. The blood serum levels of 11 cytokines were significantly different at 2 weeks post-injury when compared to uninjured controls. The levels of CXCL5, CCL11, CXCL11, IL10, TNFα, and MIF were expressed in a severity-dependent fashion, whilst CXCL1, CXCL10, CXCL11, IL2, MIP-3a, MIG, and CCL22 were expressed in a manner dependent upon the region of injury. Our pilot study demonstrates the value of using blood serum concentrations of cytokines as a rapid and affordable means of accurately classifying SCI severity in patients, removing the risks and complications associated with a reliance on repeated CSF sampling.

## Figures and Tables

**Figure 1 brainsci-11-00322-f001:**
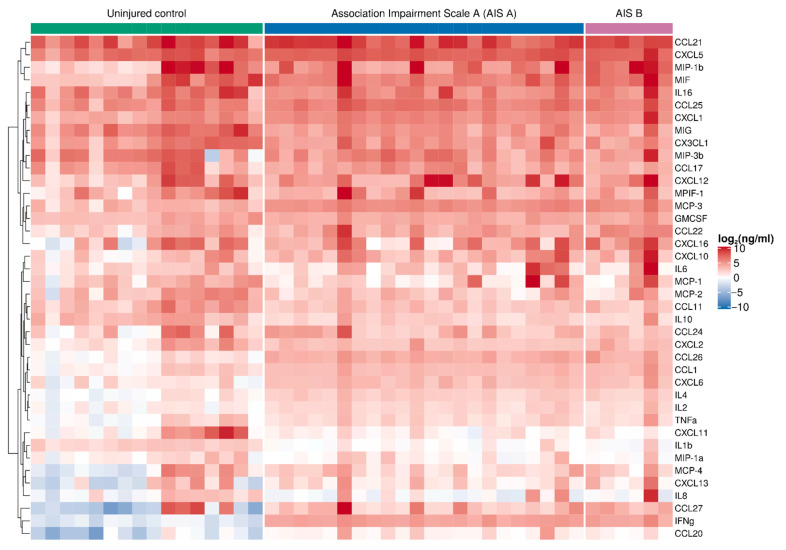
Graphical representation showing log2 cytokine concentrations (color keys), generated with the multiplex analysis of the blood serum collected at 2 weeks post-injury (*n* = 28) or from uninjured controls (*n* = 16). A dendrogram resulting from hierarchical clustering of cytokines is shown on the left.

**Figure 2 brainsci-11-00322-f002:**
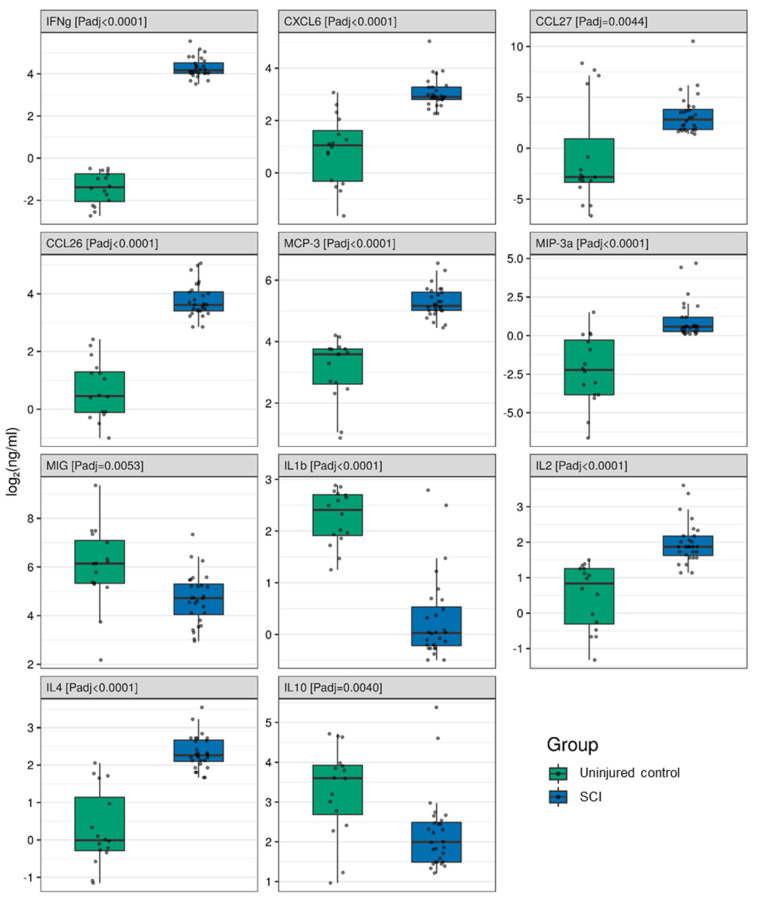
Log2-transformed blood serum cytokine concentrations (ng/mL) between 2 weeks post-spinal cord injury patients (blue columns, *n* = 28) and uninjured subjects (green columns, *n* = 16).

**Figure 3 brainsci-11-00322-f003:**
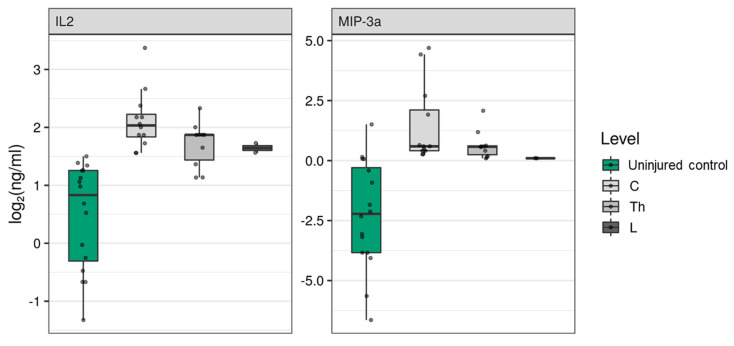
Log2-transformed blood serum cytokine concentrations (ng/mL) between patients at 2 weeks post-spinal cord injury (gray columns, *n* = 28), considering the cohort of cervical (C, *n* = 13), thoracic (Th, *n* = 12), and lumbar (L, *n* = 3) patients, and uninjured control subjects (green columns, *n* = 16).

**Table 1 brainsci-11-00322-t001:** Patient demographics and neurological status in patients 2 weeks post-spinal cord injury.

Characteristics	AIS Grade A	AIS Grade B
No. of subjects	22	6
Gender (Male/Female)	16/6	3/3
Age	40.2 ± 13.9	32.5 ± 6.3
Etiology		
traffic accident	10	1
fall	10	3
other	2	2
Regions of lesion		
cervical	13	0
thoracic	9	3
lumbar	0	3

AIS—American Spinal Injury Association Impairment Scale.

**Table 2 brainsci-11-00322-t002:** Concurrent conditions and medication history in spinal cord injury (SCI) patients 2 weeks post-injury.

Comorbid and Proinflammatory Conditions Numbers		Concurrent Medications Numbers	
Urinary tract infections Interstitial cystitis	0 1 (AIS A)	Intensive therapy * anti-inflammatory medications Dexamethasone 4.0 i/m № 7 (3 days)	28 (22 AIS A, 6 AIS B)
Community-acquired pneumonia	1 (AIS A)	spasmolytic drugs Spasmalin 5.0 i/m once a day (5 days) Tizanidine 12 mg once a day (10 days)	5 (AIS A) 1 (AIS A)
Post-traumatic pneumonia	2 (AIS A)	analgesics Ketorol 1.0 i/m once a day (5 days) Tramadol 50 i/m once a day (2 days)	23 (17 AIS A, 6 AIS B) 5 (AIS A)
Spondyloarthrosis	1 (AIS A)	Supportive therapy ** Anticoagulants Enixum 0.4 once a day	24 (20 AIS A, 4 AIS B)
Smoker	5 (4 AIS A, 1 AIS B)	Antibiotics Levofloxacin 500 mg once a day Ceftriaxone 2.0 once a day	1 (AIS A) 2 (AIS A)
Hypertensive heart disease	1 (AIS B)	Hypertension Amlodipine 5 mg once a day	1 (AIS B)

* Intensive therapy is generally carried out during the first week after SCI. ** History data collected within 5 days before blood serum sampling. AIS—American Spinal Injury Association Impairment Scale. Groups of drugs are indicated in italics.

**Table 3 brainsci-11-00322-t003:** Cytokine concentrations (ng/mL) in blood serum at 2 weeks post-spinal cord injury at cervical (C, *n* = 13), thoracic (Th, *n* = 12), and lumbar (L, *n* = 3) regions in patients and uninjured control subjects (*n* = 16).

Markers	Uninjured Control	C	Th	L
CXCL1	26.80 (20.21)	47.96 (18.06)	39.12 (13.17)	31.54 (3.55)
	18.16 (13.23‒40.48)	46.08 (34.55‒59.12) *	34.23 (29.90‒45.37)	31.54 (30.29‒32.80)
CXCL10	10.76 (17.69)	26.58 (31.55)	15.16 (14.05)	5.10 (0.13)
	4.26 (1.79‒9.72)	15.92 (6.98‒26.43) *	9.36 (5.93‒19.52)	5.10 (5.05‒5.14)
CXCL11	72.70 (175.50)	2.01 (1.62)	2.39 (1.08)	1.00 (0.02)
	2.02 (1.21‒36.61)	1.52 (1.11‒1.97) *	2.12 (1.80‒2.71)	1.00 (1.00‒1.01)
MIG	114.44 (154.44)	31.74 (19.65)	27.13 (11.64)	15.27 (7.63)
	70.53 (40.30‒136.70)	27.13 (16.55‒44.95) *	26.45 (18.75‒36.73) *	15.27 (12.57‒17.96)
CCL22	10.41 (9.80)	59.53 (149.58)	24.67 (17.01)	47.52 (2.55)
	7.10 (3.43‒14.98)	15.77 (7.56‒20.01)	16.87 (10.59‒37.67) *	47.52 (46.62‒48.42) *
CCL26	2.00 (1.46)	15.30 (6.07)	14.54 (7.50)	8.29 (1.51)
	1.38 (0.92‒2.46)	13.19 (11.11‒16.84) #	12.02 (10.81‒14.72) #	8.29 (7.75‒8.82) **
CXCL6	2.61 (2.25)	8.88 (3.19)	8.01 (2.42)	6.60 (0.54)
	2.07 (0.80‒3.11)	7.58 (7.47‒10.11) #	7.22 (6.98‒8.62) #	6.60 (6.40‒6.79) *
CCL1	3.25 (2.62)	9.63 (3.47)	7.88 (0.87)	7.30 (0.40)
	1.90 (1.40‒4.96)	8.55 (7.78‒9.53) #	7.92 (7.58‒8.36) #	7.30 (7.16‒7.44) *
IFNγ	0.42 (0.20)	20.44 (6.00)	20.13 (6.46)	13.86 (3.42)
	0.39 (0.24‒0.60)	18.98 (17.10‒21.03) #	17.38 (16.27‒23.45) #	13.86 (12.65‒15.06) #
IL10	12.40 (7.81)	6.41 (5.85)	3.77 (1.43)	2.71 (0.13)
	12.13 (6.47‒15.23)	4.97 (3.79‒6.02) *	3.27 (2.74‒4.99) **	2.71 (2.67‒2.76) *
IL1b	5.08 (1.67)	1.69 (1.37)	1.09 (0.46)	0.77 (0.08)
	5.32 (3.77‒6.51)	1.16 (1.00‒1.68) #	0.93 (0.87‒1.02) #	0.77 (0.74‒0.80) #
IL4	1.58 (1.21)	5.87 (2.12)	4.66 (1.18)	3.93 (0.61)
	0.99 (0.82‒2.25)	4.87 (4.55‒6.60) #	4.75 (3.81‒5.25) #	3.93 (3.71‒4.15) **
MCP-3	10.28 (5.16)	42.35 (14.34)	37.33 (9.59)	27.09 (7.38)
	12.02 (6.15‒13.56)	35.93 (34.13‒50.99) #	37.09 (30.44‒44.48) #	27.09 (24.48‒29.70) *

* P_adj_ < 0.05, ** P_adj_ < 0.01, and # P_adj_ < 0.0001 comparing to uninjured control subjects.

**Table 4 brainsci-11-00322-t004:** Neuron-specific enolase (NSE) and vascular endothelial growth factor (VEGF) concentrations (pg/mL) in blood serum in 2 weeks post-spinal cord injury patients and uninjured control subjects.

Markers	Uninjured Control	SCI	AIS A	AIS B
NSE	1.54 (0.80)	3.31 (3.15)	2.85 (2.51)	4.74 (4.61)
	1.40 (1.10‒1.62)	2.10 (1.50‒3.70) *	2.05 (1.52‒3.38)	3.00 (1.50‒6.20)
VEGF	233.10 (220.59)	373.03 (302.61)	368.27 (282.24)	368.27 (282.24)
	179.54 (75.84‒356.80)	324.08 (107.85‒592.85)	344.73 (111.05‒587.15)	344.73 (111.05‒587.15)

* P_adj_ < 0.05 comparing to uninjured control subjects. AIS - American Spinal Injury Association Impairment Scale.

## Data Availability

The data presented in this study are available on request from the corresponding author. The data are not publicly available due to the evolving nature of the project.

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
