# Peer review of "Blood Serum Cytokines in Patients with Subacute Spinal Cord Injury: A Pilot Study to Search for Biomarkers of Injury Severity"

_brainsci, 2021, doi:10.3390/brainsci11030322_

Round 1

Reviewer 1 Report

general comments

This is an important subject and would merit a more critical and analytical approach to the data presented.

Data presentation is incomplete, often unclear and, in parts, it is misleading and some of the overall conclusions are inaccurate.

for example, important considerations relating to the limitations of the sample protocol and controls used is not discussed at all

also, a substantial proportion of data presented as significantly different show substantial overlap of means and ranges

there is duplication in data presented in different figures

the methods section is incomplete, for example, the timing of patient sample collection indicated directly contradicts a later statement that samples were withdrawn at 1,2 and 3 weeks post trauma

the results section does not indicate timings or number of samples

information presented in results, legends, discussion sections are misplaced

the abstract does not refer to any quantitative information defining the parameters of the study, the changes in cyotokine concentrations, number of patients studied

Specific Comments

Title: is unclear. Omit 'and level'

Abstract: lacks any quantitative information, should cite number of patients, controls, age, timing of sampling, significance level of changes, concentrations of more significant patient/control cytokines.

Introduction: see Discussion, a whole introductory section justifying the study has been placed in the discussion section and should be transferred to give the study its proper context.

p1 tract delete s

p2 para1 sentence1 too long and convoluted, end at 'practice'. insert 'such studies are..necessary for'..application

para 2 VEGF is not an 'inflammatory protein'

materials and methods: this section is very incomplete

it is unclear what number of samples were analysed, and the timing of sampling is unclear

very little detail of controls is given, were these taken at the same time as patient samples? were patient and control samples analysed together in the same assay? were control/patient samples blinded? what were the therapeutic regimes of patients/controls? timing of sampling is unclear.

given that secondary inflammatory markers are being analysed, it is of absolute importance to record any anti inflammatory therapies, including dose timing, steroid treatment.

in this context, incomplete clinical profiles are presented, and additional controls are desirable, including those with unrelated inflammatory conditions, but exposed to similar therapeutic regimes to the patient group in this study.

biochemical analysis: units of measurement not given

table 1 is incomplete and should indicate any drugs given, their concentration and frequency of administration

results: timing of sampling does not concur with that given in the methods section (see major criticisms above)

there is a lack of information defining the data throughout this section, this includes details defining units of concentration, specific timings or range of times should be given, also the number of samples analysed

non standard nomenclature or colloquial terms are used, such as 'heat map' rather than relating 'concentration dependence' or 'subacute period' rather than referring to exact timings after injury.

figure legends are unclear and do not adequately describe data presented

fig1 x axis healthy..insert 'controls' AIS A ..there is room to write AIS in full

y axis should be defined ass 'cytokine concentration' and add units of measurement

CXCL5,1,10,11; CCL11,22; TNFalpha do not appear to differ in median and variance

figs2 and 3 legend gives no indication of time of sampling

contain conclusions which should be in discussion section

fig3 apart from IL-2, MIP-3alpha and possibly CCL22, the other data do not appear to differ in median and variance and should not be referred to as 'clear differences' particvulatly in a figure legend, likewise fig4 where the only marginally significant result appears to be IL10, and the fig 5 data shows complete overlap of means and errors showing little difference, and this data should be tabulated

figs 4/3 and 5 duplication of data presented

results and first part of discussion section gives no information on time dependence of cytokine changes, which are important as early indicators such as IL6 usually peak within hours of trauma

discussion: this section is well written, but contains introductory material justifying the study which should have been placed in the introduction section

closer comparison of published CSF/ compared with blood cytokine data should be made, together with a fuller discussion of cytokine kinetics and controls [see overall conclusions above]

the conclusions are not fully justified [see results section analysis above]

'master' regulator..it is more usual to refer to 'key' or 'upstream' regulators

Author Response

This is an important subject and would merit a more critical and analytical approach to the data presented.

Authors: We would like to thank the reviewer for highlighting the importance of our study and steps for improvement. We hope that our next responses will satisfy the reviewer. Our revision is visible, using the "Track Changes" function in Microsoft Word.

Data presentation is incomplete, often unclear and, in parts, it is misleading and some of the overall conclusions are inaccurate. for example, important considerations relating to the limitations of the sample protocol and controls used is not discussed at all

Authors: We have added the study limitations in Manuscript and discussed how these impact on the interpretation of the results. We have also sought to address any areas that were unclear or misleading.

there is duplication in data presented in different figures. figs 4/3 and 5 duplication of data presented

Authors: The same analytes are used in fig.2-4. However, this is due to the need to view the data based on segregation of the AIS group and SCI level. Therefore, it would not be considered duplication, as it captures the data and presents it specifically to address two different research questions. Such a presentation of the obtained data, in our opinion, makes it possible to better capture the existing differences.

the methods section is incomplete, for example, the timing of patient sample collection indicated directly contradicts a later statement that samples were withdrawn at 1,2 and 3 weeks post trauma

Authors: We apologize for our omission. We have now included the assessment data of AIS grade at weeks 1 to 3 post injury. As well as details on the timing and method of patient and control sample collection which for patients was always at 2 weeks post injury. We have corrected this contradiction in Manuscript.

the results section does not indicate timings or number of samples

Authors: We have indicated timings or number of samples where it is needed. Inserting this detail at key points as well as within the figure for ease of the reader. We have tried to avoid overloading each sentence with these same details to circumvent excessive repetition.

information presented in results, legends, discussion sections are misplaced

Authors: We have improved the layout of the manuscript as well as increasing the detail in the figure legends, results and discussion sections to address this point.

the abstract does not refer to any quantitative information defining the parameters of the study, the changes in cyotokine concentrations, number of patients studied

Authors: We have added new information in our abstract. However, unfortunately, we cannot present the obtained quantitative data due to the large number of detected analytes tested and word limit this would use, but we have detailed the key findings.

Specific Comments

Title: is unclear. Omit 'and level'

Authors: According to the reviewer’s comment, we have omit 'level' in Title.

Abstract: lacks any quantitative information, should cite number of patients, controls, age, timing of sampling, significance level of changes, concentrations of more significant patient/control cytokines.

Authors: As pointed out above, we have added more detail of the context of the study and included the key findings. To detail all of the methodological data and significant findings would expand the abstract size considerable. We felt displaying the conclusions within the abstract provides the reader with a clearer overview of the findings of the study.

Introduction: see Discussion, a whole introductory section justifying the study has been placed in the discussion section and should be transferred to give the study its proper context.

Authors: This has been moved.

p1 tract delete s

Authors: If we understood correctly, then it is done.

p2 para1 sentence1 too long and convoluted, end at 'practice'. insert 'such studies are..necessary for'..application

Authors: We rewrote relevant sentences: «Considering that the biological and physiological aspects of SCI cannot always be compared between animal models and the injured human spinal cord, there is often a lack of translation of the results of these studies into clinical practice [12]. Such studies are necessary for the application of new experimental methods for the treatment of neuroinflammation.»

para 2 VEGF is not an 'inflammatory protein'

Authors: We have added clarity to the above point.

materials and methods: this section is very incomplete

it is unclear what number of samples were analysed, and the timing of sampling is unclear. very little detail of controls is given, were these taken at the same time as patient samples? were patient and control samples analysed together in the same assay? were control/patient samples blinded? what were the therapeutic regimes of patients/controls? timing of sampling is unclear.

Authors: According to the reviewer’s comment, we have added more complete information in the Materials and Methods whilst still trying to avoid excessive duplication (3.1. Subject Demographics). The Results (3.1. part) were also expanded and supplemented.

given that secondary inflammatory markers are being analysed, it is of absolute importance to record any anti inflammatory therapies, including dose timing, steroid treatment.

Authors: We agree with the reviewer’s comment and have added this information by the incorporation of a new table (Table 2) with the corresponding data.

in this context, incomplete clinical profiles are presented, and additional controls are desirable, including those with unrelated inflammatory conditions, but exposed to similar therapeutic regimes to the patient group in this study.

Authors: The aim of this work is to identify the features of the cytokine profile of patients with SCI in the subacute period (2 weeks post-injury). At the same time, we agree with the reviewer’s comment that various therapeutic regimes taken by SCI patients can affect the results obtained and preventing an entirely objective picture being presented. However, finding non-SCI patients exposed to the same treatment regimes would be a very difficult group to obtain and deliberate exposure of healthy people without SCI to the same  drugs is not possible. In addition, it should be noted that intensive therapy (anti-inflammatory medications, spasmolytic drugs and analgesics) was carried out during the first week after SCI. We did not obtain blood samples until 2 weeks post-injury, during this period intensive medication support is not given.

biochemical analysis: units of measurement not given

Authors: We indicated units of measurement in Manuscript (Results, suppl.fig S1, figures).

table 1 is incomplete and should indicate any drugs given, their concentration and frequency of administration

Authors: According to the reviewer’s comment, we have added Table 2 with relevant information.

results: timing of sampling does not concur with that given in the methods section (see major criticisms above). there is a lack of information defining the data throughout this section, this includes details defining units of concentration, specific timings or range of times should be given, also the number of samples analysed.

Authors: This has now being corrected through the inclusion of additional methodological and experimental details.

non standard nomenclature or colloquial terms are used, such as 'heat map' rather than relating 'concentration dependence' or 'subacute period' rather than referring to exact timings after injury. figure legends are unclear and do not adequately describe data presented. fig1 x axis healthy..insert 'controls' AIS A ..there is room to write AIS in full. y axis should be defined ass 'cytokine concentration' and add units of measurement.

Authors: According to the reviewer’s comment, we have revised all terms and fig. legends.

figs2 and 3 legend gives no indication of time of sampling

Authors: Done.

contain conclusions which should be in discussion section

Authors: Done.

fig3 apart from IL-2, MIP-3alpha and possibly CCL22, the other data do not appear to differ in median and variance and should not be referred to as 'clear differences' particvulatly in a figure legend, likewise fig4 where the only marginally significant result appears to be IL10, and the fig 5 data shows complete overlap of means and errors showing little difference, and this data should be tabulated. CXCL5,1,10,11; CCL11,22; TNFalpha do not appear to differ in median and variance

also, a substantial proportion of data presented as significantly different show substantial overlap of means and ranges

Authors: In our study we used robust rank tests (1) and represented sample distributions with boxplots (2). (1) Preference of rank tests in comparing two or more groups is a well accepted option of analysis [https://hbiostat.org/doc/bbr.pdf , P 206–]. Rank tests do not imply null hypothesis in form of µ12=...=µn. The most general null hypothesis specification in the case of 2 or more groups is the absence of any difference between the distribution functions. However, in our opinion the most valuable specification for such type of hypotheses testing is P(Xi < Yj) = ½ where Xi and Yare random variables representing data generation process for two groups (this can be extended to N groups) [https://books.google.ru/books?id=ZPXRBQAAQBAJ&printsec=frontcover&source=gbs_atb&redir_esc=y#v=onepage&q&f=false P 5]. We think that visual representations of sample data are completely consistent with the results of testing procedures as described above and concordant with similar data published in the field. We do accept and agree that the differences of “centers” of sample distributions for some cytokines do not look hugely different on a log2 scale (rank tests are invariant to monotonic transformations).

(2) In our sample distributions visualizations we used “classical” boxplots proposed by Tukey [https://doi.org/10.2307/2683468], they do not include means and errors. We have to emphasize that in all plots the cytokines concentrations were log2-transformed (this transformation reduces the size between groups and individual observations within groups).

results and first part of discussion section gives no information on time dependence of cytokine changes, which are important as early indicators such as IL6 usually peak within hours of trauma

Authors: Done.

discussion: this section is well written, but contains introductory material justifying the study which should have been placed in the introduction section

Authors: Done.

closer comparison of published CSF/ compared with blood cytokine data should be made, together with a fuller discussion of cytokine kinetics and controls [see overall conclusions above]

Authors: We have added additional information in Discussion.

the conclusions are not fully justified [see results section analysis above]

Authors: We have revised conclusion.

'master' regulator..it is more usual to refer to 'key' or 'upstream' regulators

Authors: Done.

Reviewer 2 Report

Thank you for giving me the opportunity to review this paper. I think there is significant merit in the study. I believe the authors have very clearly stated their findings, without trying to over-interpret them either. As such the results can be assimilated by your readers, and added to their own perspectives.

Whilst my work has had more on brain injury, there is no doubt that the neuroinflammatory reaction is a critical determinant of patient outcome. Unfortunately, that inflammatory response is very complex - it varies between patients, it varies with the nature of the injury, and it varies with time. However, we can say that there is that relationship with patient outcome. The better we can understand the inflammatory response, then the more chance we have of translating novel anti-inflammatory agents into clinical practice. As such, I endorse the publication of this paper.

I noticed just a few English / typographical issues:

Line 47 - "..lack of translational of these studies.." should probably be something like "...there is often a lack of translation of the results of these studies into clinical practice..."

Line 363 - Ref 37 Authors should be Schwartz K, Henzel MK, Richmond MA....  

Author Response

Thank you for giving me the opportunity to review this paper. I think there is significant merit in the study. I believe the authors have very clearly stated their findings, without trying to over-interpret them either. As such the results can be assimilated by your readers, and added to their own perspectives.

Whilst my work has had more on brain injury, there is no doubt that the neuroinflammatory reaction is a critical determinant of patient outcome. Unfortunately, that inflammatory response is very complex - it varies between patients, it varies with the nature of the injury, and it varies with time. However, we can say that there is that relationship with patient outcome. The better we can understand the inflammatory response, then the more chance we have of translating novel anti-inflammatory agents into clinical practice. As such, I endorse the publication of this paper.

Authors: We would like to thank the reviewer for this review and approval the publication of this paper.

I noticed just a few English / typographical issues:

Line 47 - "..lack of translational of these studies.." should probably be something like "...there is often a lack of translation of the results of these studies into clinical practice..."

Authors: We agree with the reviewer’s comment and rewrote the relevant sentences.

Line 363 - Ref 37 Authors should be Schwartz K, Henzel MK, Richmond MA.... 

Authors: Done.

Round 2

Reviewer 1 Report

General Comments

Many of the recommended changes have not been carried out, and I cannot recognize the improvement of this manuscript in its current form, as many of the claims made are misleading,

the abstract is still lacking in both qualitative and quantitative analysis and fails to report the most significant changes in SCI, for which reasonable patient numbers are available ie patient/control groups. This information is obscured by non significant changes, and for example, the extent of the substantial difference in IFNg is not highlighted in the abstract

Throughout the manuscript, data whose distributions substantially overlap (CCL11,22,25, TNFa, CXCL1,11,5,10, VEGF) are unnecessarily presented (Figs 2-5), obscuring information about cytokines which appear to show changes which may be significantly changed in SCI.

As requested previously, overlapping and less marked differences should be presented as tables

data presentation is unclear, as mentioned previously, there is unnecessary replication of data, and figures contain unnecessary data which is marginal significance, which should be presented in table form.

Only for the more significant data, which should only include information of groups for which patient number is>6,

the distribution of individual patient results should be plotted, with clear indications of number of patients in each group, described in the legend

In the more significant groups, individual patient data should be presented, with the number of patients in each group defined in legends.

An example of the unnecessary exaggeration of small differences in the means of overlapping data can be found in the results section, p4-6: it is inaccurate to suggest that 100% of the 40 cytokines analysed showed significant changes when comparing SCI/control.

It was previously requested that the kinetics of the changes should be discussed, but no justification of the single time point used in this study can be found in the introduction, methods or discussion sections.

As mentioned previously, Data groups of n=3 are not acceptable (fig3)

Even though this subject was raised previously, figure legends still include sections that are part of the discussion section

Even though this was mentioned in the previous referee report of this manuscript, Figures still contain duplicate data are essentially unchanged, despite requests to present some of the less significant data in tablulated format.

Specific Comments

fig1 still omits the units of cytokines concentration previously requested, still omits the heading for cytokines and clarification of axes, as previously requested, please refer back to previous referee report.

fig2 individual patient and control cytokines results should be displayed, and as previously requested, the cytokines showing significant overlap ie CXCL1,11,5,10; CCL1,11,5, TNFa should be tabulated and not included in this over-large and unclear figure. Most of the legend still includes extensive discussion in legend, despite previous referee request. This should be removed. ‘Subacute’ see previous request about nonstandard nomenclature. Only IFNg, IL1b,2,4,10; CXCL6,26,27, MCP3,MIP3a,MIG should be included in this figure.

figs3-5 subgroup analysis is repetitive and should be presented as tables

fig3 as previously discussed, only MIP3a and IL2 should be included, others should be tabulated, patient numbers in each group should be indicated in legend, which should not include discussion

fig4 this data should be tabulated

fig5 this data should be tabulated

As requested previously, legends should not include discussion

title: as previously suggested, delete ‘and level’

abstract: as previously mentioned, no quantitative changes (or indeed qualitative changes) are described in cytokines.

As previously mentioned, significance is claimed where means substantially overlap. 

As previously requested, quantitative and qualitative differences should be more closely analysed in both results and discussion sections

English usage: it is surprising that, with a British co author, basic English usage is erratic in parts of this manuscript. It is requested that this collaborator should be circulated with the second revision of the manuscript, and that she should be asked specifically to address this aspect and to be fully involved and be requested to correct English usage fully before the second resubmission of this manuscript, as all authors are responsible for standards of presentation.

Author Response

General Comments

Many of the recommended changes have not been carried out, and I cannot recognize the improvement of this manuscript in its current form, as many of the claims made are misleading,

the abstract is still lacking in both qualitative and quantitative analysis and fails to report the most significant changes in SCI, for which reasonable patient numbers are available ie patient/control groups. This information is obscured by non significant changes, and for example, the extent of the substantial difference in IFNg is not highlighted in the abstract

Authors: We have added in quantitative data for IFNg, CCL27, CCL36, IL2, MIP-3a to improve the abstract and highlight those findings of substantial differences.

Throughout the manuscript, data whose distributions substantially overlap (CCL11,22,25, TNFa, CXCL1,11,5,10, VEGF) are unnecessarily presented (Figs 2-5), obscuring information about cytokines which appear to show changes which may be significantly changed in SCI.

As requested previously, overlapping and less marked differences should be presented as tables

Authors: Done.

data presentation is unclear, as mentioned previously, there is unnecessary replication of data, and figures contain unnecessary data which is marginal significance, which should be presented in table form. Only for the more significant data, which should only include information of groups for which patient number is>6, the distribution of individual patient results should be plotted, with clear indications of number of patients in each group, described in the legend. In the more significant groups, individual patient data should be presented, with the number of patients in each group defined in legends.

Authors: Done.

An example of the unnecessary exaggeration of small differences in the means of overlapping data can be found in the results section, p4-6: it is inaccurate to suggest that 100% of the 40 cytokines analysed showed significant changes when comparing SCI/control.

Authors: We have deleted inaccurate statements about significant differences.

It was previously requested that the kinetics of the changes should be discussed, but no justification of the single time point used in this study can be found in the introduction, methods or discussion sections.

Authors: We agree with reviewer’s comment and consider that using the single time point is limitation of study which was reflected in the corresponding section.

As mentioned previously, Data groups of n=3 are not acceptable (fig3)

Authors: We agree with reviewer’s comment that n=3 in lumbar level SCI group is small sample. Nevertheless, we believe that this data should not be deleted, as it may still provide important insights to those within the field for further study. We have taken care to indicate the number of patients in each group, so this is clear and we indicated the low numbers within the study limitations section.

Even though this subject was raised previously, figure legends still include sections that are part of the discussion section

Authors: Extensive discussion within legend was deleted.

Even though this was mentioned in the previous referee report of this manuscript, Figures still contain duplicate data are essentially unchanged, despite requests to present some of the less significant data in tablulated format.

Authors: According to the reviewer’s comment, less significant data were tabulated.

Specific Comments

fig1 still omits the units of cytokines concentration previously requested, still omits the heading for cytokines and clarification of axes, as previously requested, please refer back to previous referee report.

Authors: Done.

fig2 individual patient and control cytokines results should be displayed, and as previously requested, the cytokines showing significant overlap ie CXCL1,11,5,10; CCL1,11,5, TNFa should be tabulated and not included in this over-large and unclear figure. Most of the legend still includes extensive discussion in legend, despite previous referee request. This should be removed. ‘Subacute’ see previous request about nonstandard nomenclature. Only IFNg, IL1b,2,4,10; CXCL6,26,27, MCP3, MIP3a, MIG should be included in this figure.

Authors: According to the reviewer’s comment, we have made the following changes to figure 2:

-           only IFNg, IL1b,2,4,10; CXCL6,26,27, MCP3, MIP3a, MIG were included in this figure;

-           extensive discussion in legend was deleted;

-           individual patient and control cytokines results were displayed in figure;

-           nonstandard nomenclature (‘Subacute’) was deleted from figure legend.

CXCL1,11,5,10; CCL1,11,5, TNFa cytokines were also tabulated and not included in this figure (see Supplementary Table S1).

fig3 as previously discussed, only MIP3a and IL2 should be included, others should be tabulated, patient numbers in each group should be indicated in legend, which should not include discussion

Authors: According to the reviewer’s comment, we have made next changes in the figure 3:

-           only MIP3a and IL2 were included in this figure;

-           extensive discussion in legend was deleted;

-           individual patient and control cytokines results were displayed in figure;

Other cytokines were also tabulated and not included in this figure (see Table 3).

fig4 this data should be tabulated

fig5 this data should be tabulated

Authors: Fig.4 data were added in Supplementary Table S1. fig5 data were tabulated (see Table 4).

As requested previously, legends should not include discussion

Authors: Done.

title: as previously suggested, delete ‘and level’

Authors: Done.

abstract: as previously mentioned, no quantitative changes (or indeed qualitative changes) are described in cytokines.

As previously mentioned, significance is claimed where means substantially overlap. 

As previously requested, quantitative and qualitative differences should be more closely analysed in both results and discussion sections

Authors: We have deleted ‘significance’ in case than means substantially overlap.

English usage: it is surprising that, with a British co author, basic English usage is erratic in parts of this manuscript. It is requested that this collaborator should be circulated with the second revision of the manuscript, and that she should be asked specifically to address this aspect and to be fully involved and be requested to correct English usage fully before the second resubmission of this manuscript, as all authors are responsible for standards of presentation.

Authors: The manuscript has been corrected each revision version by a proficient English speaker, our colleague

Round 3

Reviewer 1 Report

There are fundamental problems with data analysis and presentation in this manuscript, which need to be fully addressed before it could be recognized.

There are defects in study design. Data groups being compared are often widely different in size (fig3), and even the overall control group (n=16) does not reach the size of the SCI patient group (n=28).

As previously pointed out, only statistically significant data should be described in results and discussion.

Previously requested data on patient clinical history is still not supplied.

Headings and legends are remarkably uniformative and should not contain redundant terminology, as previously requested. Legends should fully describe patient numbers, as previously requested.

A substantial concern is the significance of subgroup analysis, especially when a patient number of only 3 is quoted. This has been raised previously with the authors.

Finally, there may be ethical concerns about gift authorship in this paper. As previously raised, all authors should be responsible for the full content of the paper, however, despite specific requests that the UK author be involved in previous drafts, this has not been done.

Specific Comments

Headings and legends: it was requested that the UK author take responsibility for the standard of scientific English used, but, despite this, unclear and uninformative headings, redundant or ambiguous terminology continues unchanged, and vital information is omitted from legends and headings.

e.g.

Results headings 3.2, 3.3,3.4 add ‘cytokines’ to title

Results 3.3 heading: the meaning of this heading is opaque.

table 3 legend: as previously requested, n of patients in each group should be specified. Also, as previously pointed out repeatedly, the term ‘levels’ is redundant when the term ‘concentration’ is present or implied in the same sentence, although it is possible that the term ‘level’ may refer in this context to the injury subgroup, in which case a more appropriate term should be used. 

Methods: Table 2: as previously raised, agents such as anti-inflammatory drugs impact potentially on the subsequent cytokine response of patients, and full details of the dose and frequency of administration of these drugs during the initial stages of intensive care should be fully described, both in the total SCI group in this table and in subsequent sub groups.

Results: Despite repeated requests to amend this, the results section begins with statements about the patient and control groups which are misleading and are difficult for the reader to verify, suggesting that 100% of cytokines in the SCI group differed from controls (12 decreased 28 increased,in a cytokine group of 40). As previously requested, only significant differences between patient and control cytokine concentrations should be referred to.

P4 l. 145 delete sentence about ‘28 cytokines increased’

P6 l 160 delete reference to ‘12 cytokines’ ‘decreased’ 

Author Response

There are defects in study design. Data groups being compared are often widely different in size (fig3), and even the overall control group (n=16) does not reach the size of the SCI patient group (n=28).

Authors: this is a pilot study and as such the limitations of the study design have been clearly highlighted within the manuscript (see study limitation section within the discussion). Despite being a pilot study we have identified findings that are of interest to the field, as highlighted positively by the other reviewers of our paper.

As previously pointed out, only statistically significant data should be described in results and discussion.

Authors: We have made clear where data is and is not statistically significant. As this is a pilot study, we felt it important to illustrate all of our findings for the benefit of the field and future larger scale studies.

Previously requested data on patient clinical history is still not supplied.

Authors: We disagree with the reviewer. This data on patient clinical history was supplied in the first request (see Table 2). If the reviewer requires further clinical data, please can they specify the exact information that they are requesting.

Headings and legends are remarkably uniformative and should not contain redundant terminology, as previously requested. Legends should fully describe patient numbers, as previously requested.

Authors: We have added in details of patient numbers and reviewed the figure legends. In line with scientific writing convention that legends should enable figures to be stand alone items, we feel our descriptions are informative and appropriate. If the reviewer still feels this is not the case, perhaps they can highlight the occasions where we have missed the mark on this.

A substantial concern is the significance of subgroup analysis, especially when a patient number of only 3 is quoted. This has been raised previously with the authors.

Authors: We agree that the number of patients with lumbar injury is less then in other groups but according to all common statistical rules minimal amount of samples (n=3 in our case) is provided. For this reason, we mainly focused on patients groups with injury in other regions, but felt it was still important to report what we had observed for the lumbar group. Again, this is a pilot study and we have been very careful to make clear the numbers per groups and to illustrate the limitations of this within the manuscript.

Finally, there may be ethical concerns about gift authorship in this paper. As previously raised, all authors should be responsible for the full content of the paper, however, despite specific requests that the UK author be involved in previous drafts, this has not been done.

Authors: Authors YM, SO and VJ have been involved in the writing, drafting and editing of all versions of the manuscript both before submission and during each round of review. This is clearly stated in our author contributions section, as are all the other contributions made to this paper. As is the convention, all contact with the journal and reviewers has been made by the senior author on behalf of and with the approval of all authors involved.

Author Contributions: Conceptualization, S.O. and Y.M.; methodology, E.G.; software, M.K.; validation, Y.M., D.S. and A.K.; formal analysis, S.O., I.B., and Y.M.; investigation, S.O., I.S., I.B., E.G., D.S., A.K. and G.M.; resources, Y.M.; data curation, A.R.; writing—original draft preparation, Y.M., S.O. and V.J.; writing—review and editing, V.J.; visualization, M.K.; supervision, A.R.; project administration, Y.M.; funding acquisition, Y.M. All authors have read and agreed to the published version of the manuscript.

Specific Comments

Headings and legends: it was requested that the UK author take responsibility for the standard of scientific English used, but, despite this, unclear and uninformative headings, redundant or ambiguous terminology continues unchanged, and vital information is omitted from legends and headings.

e.g.

Authors: We feel that we have addressed all of the reviewer’s comments at each stage of the process. We have had the manuscript proof-read and as remarked on by the other reviewers, significant errors and omissions were not identified. If we have made omissions, we would request that the reviewer be more specific in identifying where these have occurred. Perhaps this is a reflection of differing writing styles between the authors and the reviewer, both of which would be acceptable in scientific and medical journals at an editors discretion.

Results headings 3.2, 3.3,3.4 add ‘cytokines’ to title

Authors: Done.

Results 3.3 heading: the meaning of this heading is opaque.

Authors: According to reviewer’s comments we proposed the term region to describe lumbar, thoracic or cervical levels of trauma.

table 3 legend: as previously requested, n of patients in each group should be specified. Also, as previously pointed out repeatedly, the term ‘levels’ is redundant when the term ‘concentration’ is present or implied in the same sentence, although it is possible that the term ‘level’ may refer in this context to the injury subgroup, in which case a more appropriate term should be used. 

 Authors: We replaced “level” to “region” to specify area of trauma. However, describing concentrations and changes of cytokine secretion term level was used, and it`s denotation doesn`t contradict to the main idea.

Methods: Table 2: as previously raised, agents such as anti-inflammatory drugs impact potentially on the subsequent cytokine response of patients, and full details of the dose and frequency of administration of these drugs during the initial stages of intensive care should be fully described, both in the total SCI group in this table and in subsequent sub groups.

 Authors: We do not think that anti-inflammatory drugs used in the first week of SCI impact potentially on the subsequent cytokine response of patients at 2 weeks post-injury. However, we added this information in Table 2.

Results: Despite repeated requests to amend this, the results section begins with statements about the patient and control groups which are misleading and are difficult for the reader to verify, suggesting that 100% of cytokines in the SCI group differed from controls (12 decreased 28 increased,in a cytokine group of 40). As previously requested, only significant differences between patient and control cytokine concentrations should be referred to.

Authors: Previously, we wrote « were increased…» and «were decreased…», but not « were significant increased…» and «were significant decreased…». To our mind, these are different notions.

However, we have changed sentences according reviewer’s comment.

P4 l. 145 delete sentence about ‘28 cytokines increased’

 Authors: Done.

P6 l 160 delete reference to ‘12 cytokines’ ‘decreased’ 

 Authors: Done.